# Detection of Urinary Molecular Marker Test in Urothelial Cell Carcinoma: A Review of Methods and Accuracy

**DOI:** 10.3390/diagnostics12112696

**Published:** 2022-11-04

**Authors:** Catalin Bulai, Petrisor Geavlete, Cosmin-Victor Ene, Isabela Bulai, Razvan-Ionut Popescu, Cristian Mares, Corina Daniela Ene, Ana-Maria Punga, Bogdan Geavlete

**Affiliations:** 1Faculty of Medicine, “Carol Davila” University of Medicine and Pharmacy, 050474 Bucharest, Romania; 2“Saint John” Emergency Clinical Hospital, 042122 Bucharest, Romania; 3Department of Animal Productions and Public Health, University of Agronomic Sciences and Veterinary Medicine of Bucharest, 011464 Bucharest, Romania; 4“Carol Davila” Clinical Hospital of Nephrology, 010731 Bucharest, Romania

**Keywords:** bladder cancer detection, molecular marker test, urothelial carcinomas, UroVysion, NMP22, FGFR3

## Abstract

Early detection of bladder cancer has a positive impact on prognosis. A variety of biomarkers have been developed to detect bladder tumors in urine early and reduce the need for cystoscopy. To detect bladder cancer, several methods are available, but their accuracy varies according to the sensitivity and specificity of each method. This review aims to highlight the established detection methods for bladder cancer based on the available literature. In addition, we aim to identify the combination of different effective detection methods that provides the highest degree of accuracy. In our study, a keyword retrieval method was used to search for appropriate English-language references. This bibliography has been indexed in PubMed and Scopus or has been found through systematic searches from 2015 to 2022. Based on an analysis of international guidelines, it has been revealed that there are numerous discrepancies and unresolved issues. The discovery of an ideal detection method for urothelial cell carcinoma biomarkers has been the subject of numerous efforts. In recent years, a wide range of off-label, experimental, novel, and combined approaches have been published on this topic. This review can contribute to the identification of accurate methods of detecting bladder cancer and highlight areas for future research that can be improved.

## 1. Introduction

On the urogenital tract, urothelial cell carcinoma (UCC) is the second most common malignancy after prostate cancer. While bladder tumors account for 90–95% of UCCs, upper urinary tracts (UUT) are relatively rare and account for only 5–10% of UCCs [1].

Many factors are likely to contribute to the severity, recurrence, or progression of bladder cancer, as well as the patient’s survival. Increasing knowledge regarding the molecular changes associated with urothelial carcinomas has led to the development of diagnostic tests, and these have been reviewed [2,3,4,5].

Both patients and urologists face the challenge of recurrent non-muscle invasive bladder cancer and subsequent cancer progression following transurethral resection [6].

As of now, patients with bladder cancer are periodically examined for cancer recurrence or progression by cystoscopy and urine cytology, the frequency of which varies depending on their individual risk factors for the disease [7]. It is widely known that cystoscopic examinations are expensive, cause considerable patient discomfort, and have variable sensitivity [8]. Moreover, urine cytology has poor sensitivity for the detection of low-grade diseases, and its accuracy is sensitive to the pathologist’s knowledge and experience [9].

Given the invasive nature of these procedures, there is a need for a less aggressive, reliable alternative. Clinical practice requires the diagnosis and assessment of patients with early bladder cancer (Bca), especially after surgery for those at high risk [10].

Numerous novel urinary biomarkers are being investigated, and six assays have already been approved by the Food and Drug Administration (FDA), but none are currently being implemented into routine clinical practice due to high costs or low sensitivity [11].

The National Comprehensive Cancer Network (NCCN) considers several markers, including UroVysion and nuclear matrix proteins (NMP22), to be useful in monitoring the recurrence of Bca. While the various guidelines highlight UroVysion’s usefulness, it is important to recognize that its application varies based on the clinical setting [12].

On the other hand, the European Association of Urology (EAU) Guidelines 2022 specify the molecular marker test UroVysion, NMP22, Fibroblast Growth Receptor (FGFR3), and microsatellite analysis in patients with negative cystoscopy to identify patients at increased risk of progression and recurrence. The use of cystoscopy, also, to detect Bca in patients with hematuria or other symptoms that suggests the presence of the disease, is the most effective method of detection. Molecular marker tests can be used as complementary procedures to cystoscopy to detect missed tumors, especially carcinoma in situ (CIS), where specificity is of particular importance [13].

Based on the worldwide guidelines for Bca, the diagnosis includes cystoscopy, biopsy, imaging methods, and urine cytology, with fluorescence in situ hybridization (FISH) and urine marker protein detection being complementary recommendations for Bca detection [14].

Urologists are faced with the challenge of devising effective, cost-effective noninvasive surveillance protocols for low-risk patients while facilitating a more proactive approach to the early detection of high-risk refractory cancers. It is necessary to conduct comprehensive research identifying new and more efficient biomarkers to facilitate diagnosis, follow-up, and screening of communities in danger. Additionally, new and more effective molecular marker tests must be introduced using improved instruments [15].

A literature search was performed using the PubMed and SCOPUS databases from 2015 to 2022. In the search query, we entered “bladder cancer detection” OR “urothelial cell carcinoma” AND “biomarkers” OR “molecular marker test” OR “UroVysion” OR “NMP22” OR “Fibroblast Growth Factor Receptor (FGFR) 3”.

As part of this review, we included studies that examine molecular markers for diagnosing bladder cancer, assessing diagnostic performance through sensitivity/specificity, positive predictive value, negative predictive value, and prevalence samples (cutoff values and methods of sampling urine such as first-voided urine, brushing urine, washing urine, and passively collecting urine).

We aim to provide an evidence-based review of the diagnosis, the molecular biomarkers, and the control of bladder cancer in this publication. This article will refer to several guidelines published by the American Urological Association (AUA), the European Association of Urology (EAU), and the National Comprehensive Cancer Network (NCCN) that provide background information and complementary recommendations for bladder cancer diagnosis.

### 1.1. Current Guidelines for Biomarkers in Urothelial Cell Carcinoma

In the surveillance of patients with Non-Muscle Invasive Bladder Cancer (NMIBC), FDA-approved several bladder tumor antigen STAT tests are available from Bard Diagnostics, Redmond, WA, USA, BTA TRAK tests, NMP 22 and NMP22 BladderChek tests from Matrix, Newton, MA, USA, ImmunoCyt tests (Diagnocure Inc., Quebec City, QC, Canada), and fluorescence in situ hybridization (FISH) tests (UroVysion Systems Vysis, Abbott Laboratories, Abbott Park, Chicago, IL, USA). Additionally, microsatellite polymorphism analysis has been recently evaluated as a test and marker [16,17]. Despite their present and future potential, the current guidelines for the management of NMIBC do not include a critical evaluation and comparison of urine-based markers [12].

In the current guidelines, some biomarkers with a strong base of evidence are not discussed due to their lack of mention in the literature. There is a proposed test for urological neoplasms that detects Nicotinamide N-Methyltransferase (NNMT) in exfoliated cells extracted from urine samples obtained from Bca patients and healthy individuals. A significant difference was observed between urine samples from patients suffering from Bca and those from controls [18]. There is no doubt that further and detailed research is needed on this marker.

The above-mentioned tests have been endorsed by the FDA for diagnosis and follow-up of the Bca along with norm techniques, but most of these are case-control studies that are conducted in populations where Bca is prevalent, so their positive predictive value is unrealistically high. Furthermore, it is difficult to read positive results from these tests when no notable findings are found upon follow-up cystoscopy. The use of these tests in everyday practice is also limited by a lack of external validation studies [12].

The past few years have witnessed the evaluation of several promising and interesting biomarkers, although only those biomarkers approved by regulatory agencies responsible for in vitro diagnostics (IVD) became commercially available biomarkers, which may be used in conjunction with cystoscopy for the primary diagnosis and follow-up of Bca. Based on diverse reports, the novel urinary biomarkers showed a higher sensitivity but lower specificity than cytology, thus not being recommended in international guidelines [14,15,19,20].

### 1.2. Biomarkers Available for Detecting Bladder Cancer according to EAU Guideline

In this section, we describe the methods available in the EAU guidelines for detecting bladder cancer, their sensitivities and specificities, as well as their advantages and disadvantages, summarized in Table 1.

#### 1.2.1. NMP22 BladderChek

Known as nuclear matrix proteins (NMPs), these proteins form part of the internal structural framework of the nucleus. These proteins help maintain the nuclear shape and organize DNA, and they play an important role in DNA replication, transcription, and gene expression [31,32,33]. Schulz et al. (2022) specified that there are two FDA-approved tests utilizing this marker, the NMP22 Bladder Cancer ELISA and NMP22 BladderChek tests. In contrast to the ELISA test, which is a quantitative test, the BladderChek test is a qualitative test performed in a laboratory and can be performed at the point of care (POC). NMP22 markers are detected in both tests by analyzing voided urine [34]. Based on various investigations and meta-analyses, the BTA Stat has a sensitivity of 64% and a specificity of 77%, while the BTA Trak has a sensitivity of 65% and a specificity of 74% [35]. According to Yafi et al., BladderChek has a sensitivity of 25% for low-grade tumors and 91% for high-grade tumors [20]. BladderChek could provide clinical insight into which patients need cystoscopy. Unfortunately, it has been found to have a high false positive rate for urinary tract infections, calculi, foreign bodies, and other genitourinary cancers [23]. A higher sensitivity was observed in the diagnosis of symptomatic patients than in the follow-up, although the specificity was similar. Despite the reduced specificity in conditions involving complement factor H, namely, in other genitourinary malignancies, benign conditions with hematuria, including lithiasis, inflammation, instrumentation, and intravesical therapy, it presented a higher sensitivity than urinary cytology [24,32].

#### 1.2.2. UroVysion/FISH (Fluorescence In Situ Hybridization)

UroVysion is a hybridization assay, using multitarget multicolor fluorescence in situ (FISH), that identifies chromosomal abnormalities in exfoliated urothelial cells [21]. According to the American Urological Association guidelines, UroVysion FISH can be used to assess the response to Bacillus Calmette-Guerin therapy and determine proper cytology. For the detection of invisible tumors, especially CIS, urinary biomarkers may be used to follow the recommendations of the EAU guidelines. Many studies have shown that urine FISH can also be used to diagnose urothelial carcinoma in patients with urinary hematuria [22,36]. FISH was examined by Yang et al. in bladder paraganglioma using urine specimens [37]. This study did not demonstrate the effectiveness of UroVysion for diagnosing rarer histological cancers. In their previous clinical work, Hu Z et al. indicated that urinary FISH can also demonstrate positive results in urachal carcinoma, which led to this research. Therefore, this study focused on the diagnostic value of FISH in patients with urachal carcinoma and compared the consistency of histological and cytological FISH results in these patients [38]. Alternatively, Nagai, T. et al. conclude in their study that UroVysion FISH, alone, is not sufficient in detecting bladder cancer, and cystoscopy needs to be accompanied by UroVysion FISH for accurate detection and follow-up [27].

#### 1.2.3. FGFR3 in Bladder Cancer

FGFR3 has been validated as a prognostic and predictive marker in urothelial bladder cancer and as a therapeutic target [39]. The presence of FGFR3 gene alterations is generally associated with a lower grade and stage of urothelial bladder carcinoma [38]. According to the American Joint Committee on Cancer’s 8th edition, FGFR3 expression is associated with a lower-grade tumor and a lower risk of cancer progression [28]. Even though FGFR3 gene alterations are generally associated with favorable characteristics, Lotan, Y et al. argue that there is no evidence that FGFR3 gene alterations correlate with an aggressive phenotype once urothelial carcinoma has advanced. When chemotherapy is administered to patients with advanced cancer, FGFR3 gene alterations are associated with less favorable outcomes [40]. Urothelial cancers are driven by FGFR3, leading to the development of therapeutics that target FGFR3 [41]. It has been shown that single-agent dovitinib, which targets FGFR3, among other tyrosine kinases, is less effective in a population of unselected urothelial cancer patients than pan-FGFR inhibitors with a higher target affinity [42,43].

#### 1.2.4. Microsatellite Detection

Microsatellite instability (MSI) is considered the diagnostic for gastrointestinal, endometrial, colorectal, and bladder cancers; however, instability events are emerging in a wider range of cancers. This process plays a crucial role in the different stages of human carcinogenesis. Microsatellite markers are presently detected through the examination of PCR products from important microsatellite markers (MSI-PCR) [29]. It has been reported that numerous groups have generated techniques for analyzing MSI using extensive parallel DNA-sequencing technologies in recent years [30]. Besides offering solid precision, both qualitative and quantitative, this recent approach is also based on a high number of samples. As well as identifying valuable biomarkers and new therapeutic targets for MSI-positive carcinogenesis, next-generation sequencing (NGS) has become a useful tool [29]. These are the latest advances in the study of MSI + (MSI positive) carcinogenesis [29,44,45,46]. As a result of the initial studies on Lynch syndrome, numerous studies of MSI for a diversity of human carcinogens have led to three key conclusions. There are two potential causes of deficiencies/impairments in the mismatch repair (MMR) system: mutational deactivation of the enzyme function or DNA methylation-based silencing of several key genes in the MMR pathway [44]. In recent research, it has been suggested that MSI may be utilized as a predictor for immune-checkpoint-blockade therapy. For the patients with MSI-positive tumors, the treatment with inhibitors of programmed cell death 1 (PD-1) has been shown to lead to better outcomes when compared to those with negative tumors. It is possible to explain this conclusion by the ability of T lymphocytes to detect multiple peptide neoantigens, produced by a variety of MSI-based DNA mutations [45].

## 2. Discussions

It seems arbitrary to compare the results of this study with those of the current specialized literature when the cut-off values as well as the types of urine sampling methods investigated were different (voided urine, urothelial brushing, urothelial washing, and passively collected urine). Comparing the analyses for the above-mentioned biomarkers, no difference in sensitivity or specificity was found between the NMP22 test kit (cut-o >10 U/mL) and the BTA Stat in different stages and tumor grades [18].

There is a wide range of sensitivity and specificity reported in the literature for FISH regarding Upper Tract Urothelial Carcinoma (UTUC) detection (35–88% and 78–96%) [47,48]. The literature also reports that urine cytology for the detection of urothelial carcinoma of the bladder and UTUC has a lower sensitivity but equal or higher specificity to that of FISH [48,49,50]. There is a need for additional research to examine the diagnostic preciseness of FISH as a triage test for ureteroscopy in the follow-up following Kidney Sparing Surgery of UTUC. As a result of our review, we conclude that urological research focuses on the early diagnosis of bladder cancer and the careful follow-up of recurrences after initial treatment. Compared to disease detection, there have been very few studies examining the use of pre-treatment urinary biomarker levels as a prognostic indicator.

The high rate of recurrences and the prolonged follow-up by cystoscopy and cytology make bladder cancer the most expensive cancer to treat overall.

In patients at risk of bladder cancer, NMP22 and UroVysion are recommended for both diagnosis and surveillance. These tests are well-studied urinary markers that may improve the diagnosis and management of new cancers by utilizing noninvasive urinary markers. Based on our review, we have found that very few studies have investigated Microsatellite Detection over 2015 although this method has been proposed by EAU.

According to the EAU Guidelines, detecting bladder cancer remains difficult due to the absence of an approved diagnostic or surveillance marker, as none of these approved markers can replace cystoscopy or extend the interval between cystoscopies [13].

The current methods of diagnosing and monitoring bladder cancer are cystoscopy and cytology, which are not considered optimal urine markers, and therefore are unable to detect bladder cancer reliably. Besides being costly and invasive, cystoscopy also lacks sensitivity for low-grade tumors. Consequently, many urine-based tests have been developed to improve efficiency beyond the current diagnostic tests. For the diagnosis and the monitoring of Bca, the EAU and FDA-approved methods (NMP22, Urovyson, FGFR3, and microsatellite detection) have demonstrated greater sensitivity but lower specificity, especially in cases of low-grade, early-stage, or recurrent Bca. In addition to the strengths of the methods listed above, invisible tumors, including carcinoma in situ, can be accurately detected with a high positive predictive value even in low-grade cases. Moreover, FGFR3 is one of the strongest correlates of prognosis that can be detected as a prognostic selection marker. The advantages of each method are balanced by disadvantages, such as higher false positive rates in urinary tract infections, the cost, and the need for standardized markers, which can limit their use in daily healthcare (e.g., UroVysion test). Moreover, bladder cancer detection and monitoring require a FISH test along with a cystoscopy. Aside from FDA-approved tests, most commercial tests are research-based.

Currently, there are several commercially available diagnostic tools to detect Bca in urine, but intensive research is in progress to discover more effective biomarkers [32,51,52]. It is likely that liquid biopsy will be a vital component of the early detection of tumors, allowing for the best possible care for patients while avoiding the human and economic costs that would result from a delayed diagnosis [53]. Due to the limited number of patients in most studies, the lack of external validation in large-scale prospective studies, the lack of comparative studies between biomarkers, and a need to methodologically improve the existing biomarkers and discover new robust biomarkers is of utmost importance. In addition, existing biomarkers in low-risk Bca are relatively underpowered and have low specificity, and they are more accurate at initial diagnosis than at follow-up [18]. The limitations of these biomarkers prevent most international clinical societies from making recommendations, and the literature suggests that single biomarkers are not sufficient to overcome these limitations. In this regard, current research trends focus on the development of more accurate diagnostic and monitoring tools for Bca, as well as on predicting its behavior to provide prognostic information [54].

There has been an insufficient assessment of tests using standard case-control methods, emphasizing the need for prospective cohort studies which involve serial samples taken from individuals at risk, at different time points, and large, randomized assays that validate the clinical benefit of biomarkers. Due to these factors, international societies’ guidelines do not recommend their use as adjuncts or substitutes to conventional cystoscopy and cytology. For now, cystoscopy and urinary cytology remain the gold standard for diagnosing bladder cancer, though further research is needed on markers that could help define their role in the disease.

There are a few limitations to be mentioned about this review to ensure its validity. In the first place, due to the publication bias in the literature, only positive data are published, so the review may be skewed in favor of more positive results. Secondly, the sensitivity and specificity of the method have not been evaluated in subgroups of tumor grade and stage, which is yet another limitation of this study.

While there are several limitations to this study, we believe that it presents a comprehensive summary of the literature and provides a searchable database of molecular markers that have been investigated so far that will be a great help to those who are looking for better urine biomarkers.

## 3. Future Directions

As of today, none of the urinary protein biomarkers investigated have been able to be used for the accurate diagnosis of bladder cancer without invasive procedures. There is currently no successful attempt to combine protein biomarkers to improve test accuracy, but neither sensitivity nor specificity is high enough to be clinically useful. A diagnostic algorithm that includes the diagnostic and follow-up protocol, as well as clinical risk stratification, could highlight the potential role of these techniques in the future for patients with suspected urothelial carcinoma. The development of technology and the collaboration between the clinical community and medical engineering departments, for example, will hopefully enable clinicians soon to perform minimally invasive biomarker tests for the detection and follow-up of bladder cancer and/or upper urothelial carcinoma patients in the near future.

## 4. Conclusions

Despite their high sensitivity and specificity, non-invasive BC assays still suffer from inconvenient rates of false positives. Bladder cancer can be detected non-invasively by a wide variety of tests, each of which has its own advantages and disadvantages. Currently, no biomarkers kit is widely used.

Due to their low specificity and limited clinical utility, urine-based biomarkers do not improve diagnostic precision, and they require more reliable, sensible, and specific urine-based biomarkers for bladder cancer.

The biomarkers described in this article cannot be recommended for widespread use in detecting, monitoring, and prognosticating bladder cancer, and future studies are expected to explore new biomarkers and targets.

We will continue this study by evaluating the recommended biomarkers to confirm the effectiveness of non-invasive bladder cancer detection methodologies and to improve patient care.

## Figures and Tables

**Table 1 diagnostics-12-02696-t001:** The results of a literature search using Scopus complete and PubMed from 2015 to 2022.

Urinary Biomarker Test	Sample Method	Assay	Purpose	Sensitivity %	Specificity %	Advantages	Disadvantages	References
NMP22 BladderChek	voided urine	SIA	Detection	85%	83%	Symptomatic patients had a higher sensitivity than follow-up patients, although the specificity was similar	Sensitivity increases with tumor size, grade, and stage. Cannot be recommended without cystoscopy due to false-positive rates.	[21,22]
			Follow-Up	25% LG-T 91% HG-T	83%	Low-grade tumor detection	False positive rates for urinary tract infections are high	[23,24]
UroVysion	urine specimens	FISH	Detection	77%	97%	Detect invisible tumors, particularly carcinoma in situ	Labor-intensive and Expensive	[19,25,26]
			Follow-Up	55%	80%	Presented a high positive predictive value	To detect and follow up on bladder cancer, FISH should be performed in addition to cystoscopy	[18,27]
FGFR3 in Bladder Cancer	urine sediment DNA samples	PCR	Detection			In bladder cancer, the FGFR3 gene is prevalent and may be useful as a prognostic marker or as a selection tool		[28]
Microsatellite Detection	DNA samples	PCR	Detection	72–97%	80–100%	Detection of recurrence or progression of bladder cancer	This clinical setting requires the use of standardized markers.	[14,29,30]

Abbreviations: SIA; sandwich immunoassay; FISH; fluorescence in situ hybridization; RT-qPCR; reverse transcription-quantitative polymerase chain reaction; SIA; sandwich immunoassay; HG-T—high-grade tumors; LG-T—for low-grade tumors.

## Data Availability

Not applicable.

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
