# Peer review of "Detection of Urinary Molecular Marker Test in Urothelial Cell Carcinoma: A Review of Methods and Accuracy"

_diagnostics, 2022, doi:10.3390/diagnostics12112696_

Round 1

Reviewer 1 Report (Previous Reviewer 2)

The manuscript has been improved and can be considered for publishing.

Author Response

Comments and Suggestions for Authors:
The manuscript has been improved and can be considered for publishing.

In response to Reviewer 1:
I would like to thank you for your valuable comments. In order to make the language clearer, we revised it.

Reviewer 2 Report (New Reviewer)

A minor revision is needed before publication.

Table 1: The reviewer wonders if "sensitivity greater than urine cytology" is a disadvantage of NMP22. 

Author Response

Comments and Suggestions for Authors:
A minor revision is needed before publication.
Table 1: The reviewer wonders if "sensitivity greater than urine cytology" is a disadvantage of NMP22.

Reply for Reviewer 2:
Thank you very much for the valuable comments.

A revision has been made to the language.

In table 1, the form has been amended to clarify the mention of the disadvantages of NMP22.

Reviewer 3 Report (New Reviewer)

On the urogenital tract, urothelial cell carcinoma (UCC) is the second most common malignancy, after prostate cancer.

Many factors are likely to contribute to the severity, recurrence, or progression of  bladder cancer, as well as the patient's survival.

Among these, one of the most relevant is definitely delayed diagnosis.

The main aim of this review is to evaluate the impact of an early diagnosis on bladder cancer in terms of survival and other.

In this regard, they focused on the possibility of using several tumor markers for early detection of bladder cancer. For this reason, they collected several studies previously done in order to evaluate the best markers that could play a pivotal role in early diagnosis (such as NMP22,FGFR3).

The results came out from this evaluation werenot satisfactory, as although these bio markers had high sensitivity and specificity, they resulted in numerous false positives.

Although the topics covered are indeed interesting, I noted the lack of insights using graphs, no less the absence of standardized cut-offs that would allow adequate and objective comparison of data.

The interesting topic discussed needs some corrections before being suitable for publication.

1. Please, check the language along the text

2. Please, at page 2 explain the abbreviation of “BCa”

3. It would be interesting to compare this review with this interesting article, considering the affinity to the topic discussed and the accuracy reported in the latter (https://doi.org/10.1016/j.critrevonc.2022.103577)

4. In the context of the importance of an early diagnosis, it would be useful to rely not only on the validity of biomarkers, but also on any instrumental investigations that might improve diagnostic accuracy. It has already been amply demonstrated in other cancers and highlighted in this article. You can find it here DOI: https://doi.org/10.3390/cancers13184723

Author Response

Comments and Suggestions for Authors:

On the urogenital tract, urothelial cell carcinoma (UCC) is the second most common malignancy, after prostate cancer.

Many factors are likely to contribute to the severity, recurrence, or progression of bladder cancer, as well as the patient's survival.

Among these, one of the most relevant is definitely delayed diagnosis.

The main aim of this review is to evaluate the impact of an early diagnosis on bladder cancer in terms of survival and other.

In this regard, they focused on the possibility of using several tumor markers for early detection of bladder cancer. For this reason, they collected several studies previously done in order to evaluate the best markers that could play a pivotal role in early diagnosis (such as NMP22, FGFR3).

The results came out from this evaluation were not satisfactory, as although these bio markers had high sensitivity and specificity, they resulted in numerous false positives.

Although the topics covered are indeed interesting, I noted the lack of insights using graphs, no less the absence of standardized cut-offs that would allow adequate and objective comparison of data.

The interesting topic discussed needs some corrections before being suitable for publication.

  1. Please, check the language along the text
    2. Please, at page 2 explain the abbreviation of “BCa”
    3. It would be interesting to compare this review with this interesting article, considering the affinity to the topic discussed and the accuracy reported in the latter (https://doi.org/10.1016/j.critrevonc.2022.103577)
  2. In the context of the importance of an early diagnosis, it would be useful to rely not only on the validity of biomarkers, but also on any instrumental investigations that might improve diagnostic accuracy. It has already been amply demonstrated in other cancers and highlighted in this article. You can find it here DOI: https://doi.org/10.3390/cancers13184723

The following is our response to Reviewer 3:

Thank you very much for the valuable comments.

  1. In order to improve the readability of the text, we revised the language.
  2. The abbreviation "BCa" on page 2 has been corrected.
  3. Based on the information you provided, we have made some mentions and references to the article (reference no 52).
  4. As you exemplified, the study you presented is very valuable and emphasizes in a very professional manner the importance of looking forward beyond the limits. Our team is totally in agreement with the concept that the use of biomarkers in conjunction with more instrumental investigations may enhance diagnostic accuracy. Our manuscript contains a reference to this, but we believe that it is a topic of further research that deserves to be studied in more detail.

Round 2

Reviewer 3 Report (New Reviewer)

Authors answered all comments and suggestions!

This manuscript is a resubmission of an earlier submission. The following is a list of the peer review reports and author responses from that submission.

Round 1

Reviewer 1 Report

Early detection of bladder cancer is important. The authors try to summarize urinary molecular marker test to detect bladder cancer. Four tests including NMP22, UroVysion, FGFR3 and Microsatellite detection were reviewed in this article. A more in-depth discussion of the data presented in the literature is necessary. Besides, the strengths and weaknesses of each test should be described more clearly. It is also due to the comparison of how these tests are recommended in the main treatment guidelines  and when to use it.

Reviewer 2 Report

The manuscript “Detection of urinary molecular marker test in urothelial cell carcinoma: A review of methods and accuracy” is a review article that analyzes the current novel options of detection methods for bladder cancer based on the available literature. The manuscript is easy to read, although few typos are present and a grammar revision might be useful. The manuscript need to be improved in order to be considered for publication, according to the following concerns:

1.       The main concern is that this manuscript completely ignored one of the proposed test for detection of bladder cancer based on NNMT detection (PMID: 34439880; PMID: 29148015). A paragraph discussing this test need to be inserted.

2.       Language revision is suggested; sometimes references are written after the dot, sometimes before. Please reconcile.

3.       In the conclusion section, authors should underline which is the most promising perspective for developing an accurate and specific test for bladder cancer. Where researchers should be focused mostly? mRNA detection? DNA? Proteins/enzymes? Molecules?